# The Art of Asking: Prompting Large Language Models for Serendipity Recommendations

## ABSTRACT

Serendipity means an unexpected but valuable discovery. Its elusive nature makes it susceptible to modeling. In this paper, we address the challenge of modeling serendipity in recommender systems using Large Language Models (LLMs), a recent breakthrough in AI technologies. We leveraged LLMs' prompting mechanisms to convert a problem of serendipity recommendations into a problem of formulating a prompt to elicit serendipity recommendations. The formulated prompt is called *SerenPrompt*. We designed three types of *SerenPrompt*: discrete with natural words, continuous with trainable tokens, and hybrid that combines the previous two types. In the meanwhile, for each type of *SerenPrompt*, we also designed two styles: direct and indirect, to investigate whether it is feasible to directly ask an LLM a question on whether an item is a serendipity, or we should breakdown the question into several sub-questions. Extensive experiments have demonstrated the effectiveness of *SerenPrompt* in generating serendipity recommendations, compared to the state-of-the-art models. The combination of the hybrid type and the indirect style achieves the best performance, with relatively low sacrifice to computational efficiency. The results demonstrate that natural words and virtual tokens, as building blocks of LLM prompts, each have their own advantages. The better performance of the indirect style speaks to the effectiveness of decomposing the direct question on serendipity.

## CCS CONCEPTS

• **Information systems** → **Recommender systems**.

## KEYWORDS

Serendipity, Large Language Models, recommendation models, prompt learning

**ACM Reference Format:**
Anonymous Author(s). 2018. The Art of Asking: Prompting Large Language Models for Serendipity Recommendations. In *Proceedings of Make sure to enter the correct conference title from your rights confirmation emai (Conference acronym 'XX).* ACM, New York, NY, USA, 10 pages. https://doi.org/XXXXXXX.XXXXXXX

## 1 INTRODUCTION

Serendipity means an unexpected but valuable discovery. As early as 1997, Gup [6] expressed his concern about the "end of serendipity" in the digital world, recalling with the fondness his childhood experiences coming across interesting tidbits of information while flipping encyclopedia pages. Gup's concerns are echoed by others in more recent studies (e.g., [18, 21, 23]). The sense that the online environment is increasingly determined promotes a widespread feeling that serendipity is threatened.

Even with today's deep learning models, modeling serendipity is difficult due to the elusive nature of serendipity. The element of unexpectedness in serendipity means surprise and accident, which are susceptible to modeling and prediction. The recent rise of Large Language Models (LLMs), especially ChatGPT, has brought global excitement about what AI could do for humans. LLMs begin taking on surprising emergent abilities when they reach a certain size. In line with Anderson's well-known suggestion that "more is different" [1], LLMs appear to go through a form of phase transition, bringing about new capacities for which they were not explicitly trained. In this paper, we investigated whether LLMs have such emergent capacities that are helpful for serendipity recommendations, a long-standing research challenge in recommender systems research community.

Since last year, initial efforts have been made to explore the potential of LLMs as a promising technique for the next generation recommender systems, due to the fact that recommender systems could be regarded as question answering (QA) systems: given a question of a user's previous preferences, the system generates an answer about this user's future preferences. In addition, recommender systems typically contain a large amount of text information, such as user reviews, item descriptions, which aligns with the data format of an LLM. Specifically, **we will leverage LLMs' prompting mechanism, the new paradigm compared to the pre-training and fine-tuning paradigm for a language model, to convert the problems of serendipity recommendations into problems of formulating prompts, to elicit serendipity recommendations.** The resulting prompts are called *SerenPrompt*. We experimented with discrete prompts with manually selected natural words, continuous prompts with trainable tokens during a lightweight model training, and hybrid prompts that combine natural words and trainable tokens.

Equally importantly, serendipity is a difficult concept to study due to its elusive and subjective nature. Most studies on serendipity have decomposed the concept into a few more tangible sub-concepts, such as diversity, novelty, coverage, unexpectedness, surprise, interestingness, value, and relevance. Very few studies have studied serendipity as a whole concept. Since LLMs are believed to be an "encyclopedia", containing comprehensive knowledge of human society through extensive pre-training, we would like to investigate whether it is feasible to directly ask an LLM a question

on whether an item is a serendipity, or we should decompose the question into several sub-questions on the sub-concepts of serendipity.

The major contribution of this paper is three-fold:

- the design of *SerenPrompt* with three types and two styles, as the first work exploring how to prompt LLMs for serendipity recommendations
- the decomposition of the task of recommending serendipity into two sub-tasks: recommending the unexpected and recommending the relevant
- a computational approach to "calculate" the ground truth of unexpectedness for the core pre-training task, avoiding a tedious human labeling process

## 2 RELATED WORK

This project draws on several research lines. We will review these areas in the following subsections.

### 2.1 The Concept of Serendipity and Its Distinction with Diversity and Novelty

The word "serendipity" is used to describe the process of making unexpected discoveries by accident. In the early 2000s, serendipity was first introduced to the context of recommender systems to broaden users' selections and increase their satisfaction [7]. While perceived as valuable, serendipity is also seen as elusive, unpredictable, and hard to control to be used [3]. Although there is some disagreement as to the precise nature of serendipity, all accounts agree that the following two aspects are central: **an unexpected chance and a relevant discovery. These two aspects have informed us on the design of** *SerenPrompt*.

It is worth distinguishing between serendipity, diversity, and novelty because they share some common characteristics. We believe diversity increases the chance of serendipity, but not every diversified piece is serendipitous: only those unexpected and relevant items are serendipity. As to novelty, it means being new and unknown, not necessarily unexpecting or surprising. In contrast, serendipity suggests how strongly an item violates an expectation. Therefore, it is worth clarifying that, different from many other studies (e.g., [2, 10, 27]), we do not use diversity and novelty as the sub-concepts of serendipity. In this paper, we interpret and operationalize the concept of serendipity using two sub-concepts: **unexpectedness and relevance**.

### 2.2 Deep Learning Models for Serendipity Recommendations

Since 2018, a few information retrieval researchers have attempted to build deep learning models for serendipity recommendations. Examples are SerRec [20], HAES [13], DESR [27], NSR [27], PURS [12], and SNPR [28]. These conventional deep learning efforts collectively demonstrate the potential of neural networks in representing users' serendipity needs. However, these studies' serendipity definitions varied in order to leverage various existing recommendation datasets and avoid collecting the direct ground truth on serendipity, making both the models and the results not sufficiently systematic or generalizable. In addition, the conventional deep learning

models' limited sequence representation capacity and limited natural language understanding capability make their performance as a recommendation model not ideal, especially for those complex and multi-step recommendation tasks including serendipity recommendations. Therefore, in this paper, we investigated the potential of LLMs as recommendation models for this long-standing challenging task of serendipity recommendations.

### 2.3 LLMs for Recommendation Models

Technically, there are three main ways to leverage LLMs for recommendation tasks: pre-training, fine-tuning, and prompting. Practically, both pre-training and fine-tuning an LLM need heavy computational resources, usually not immediately available in academia. Prompting therefore becomes the popular access to LLMs, to adapt a recommendation task into a question answering task or a language generation task, instead of the other way around during the pre-training and fine-tuning process.

Prompting is the new paradigm for adapting LLMs to specific downstream tasks. A prompt refers to a text template that serves as the input of LLMs. Prompting enables LLMs to unify different downstream tasks into language generation tasks [5]. Generally, the types of prompts can be categorized as discrete and continuous [16]. Discrete prompts consist of natural words, relying heavily on human experiences to craft. Although discrete prompts have succeeded in many tasks, handcrafted prompts may be with costs and not globally optimal. Continuous prompts introduce learnable prompt tokens to automatically search for the best prompt templates. In the following paragraph, we will briefly review the recent efforts of using the prompting techniques (both discrete and continuous) for recommendation tasks.

A straightforward prompting approach is discrete prompting. For instance, Liu et al. [15] employ ChatGPT and propose separate task descriptions with a few demonstrations (examples) tailored to different types of recommendation tasks, such as top-K recommendations, rating predictions, and explanation generation. In contrast to discrete prompts, continuous prompting employs learnable tokens (vectors or text embeddings) as a prompt. For instance, Wu et al. [26] apply contrastive learning to capture user representations and apply them into prompt tokens. In addition to directly using pre-calculated embeddings, continuous prompts can also be learned using the current task-specific loss function. For example, Zhang et al. [29] adopt randomly initialized continuous prompts and optimize them with respect to a recommendation loss function. Compared to discrete prompts, continuous prompts are more flexible for tuning on a continuous space but at the cost of explainability [8].

Those efforts mentioned above demonstrate the promising potential of prompting LLMs for recommendation tasks. All of those efforts are for the conventional accuracy-oriented recommendations. In this paper, we would like to investigate the feasibility of LLMs for serendipity-oriented recommendations, a more complex and challenging task. We believe LLMs have a huge potential for this task because of their stronger sequence representation capacity and language understanding ability.

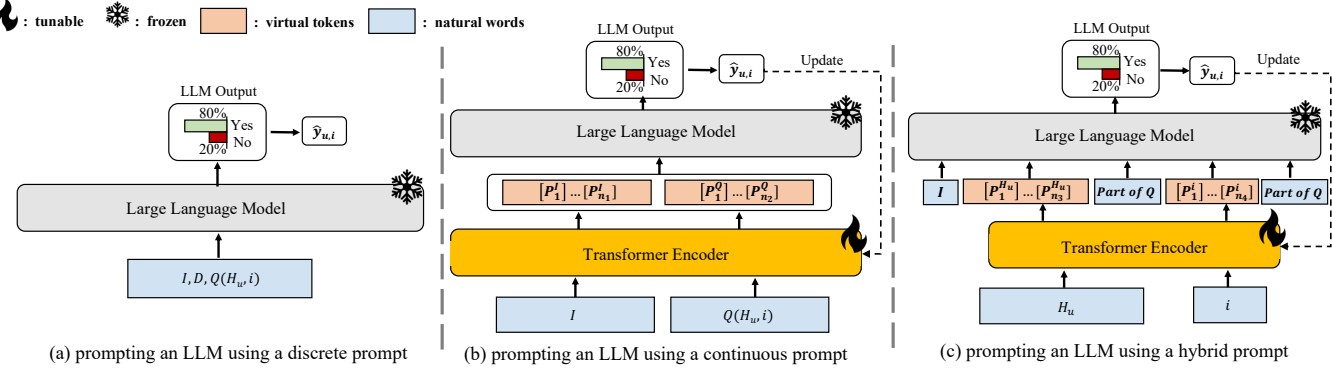

**Figure 1: Three types of prompting an LLM for serendipity recommendations**

## 3 SERENPROMPT: PROMPTING AN LLM FOR SERENDIPITY RECOMMENDATIONS

We believe a serendipity recommendation problem, like any other recommendation problem, is a matching problem. Let $I = \{i_1, i_2, \ldots, i_{|I|}\}$ represents the set of items, and $H_u = \{i_1^u, i_2^u, \ldots, i_n^u\}$ represents a history of interacted items for the user $u$. The goal is to find a model $matching(\cdot)$ to predict a matching probability score $\hat{y}_{u,i}$ that the item $i$ ($i \in I$) is a serendipity to the user $u$ with a history $H_u$. The process could be represented as:

$$\hat{y}_{u,i} = matching(H_u, i) \quad (1)$$

To convert the serendipity recommendation problem to a prompt to an LLM, we need to include the user information $H_u$ and the item information $i$ in the prompt. Commonly, a well-designed prompt for an LLM contains three parts: a task description, a few demonstrations (examples), and an input question. The task description defines a task and introduces the related concepts. The demonstrations provide some task and solution examples for the LLM to better understand the task. The input question directly asks the question. In our case, given a task description $I$, a demonstration set $D = \{(d_1, d_2, ..., d_k\}$, and the input question $Q$, which includes the current user $u$'s history $H_u$ and item $i$'s information, the prediction $\hat{y}_{u,i}$ generated from LLMs can be formulated as follows:

$$\hat{y}_{u,i} = LLM(I, D, Q(H_u, i)) \quad (2)$$

We are interested to know what kind of prompt templates benefit the performance of serendipity recommendations. To this end, **we designed three types of templates, as shown in Figure 1: 1) discrete with natural words, 2) continuous with vector tokens to search for the best prompt, and 3) hybrid that combines the discrete and continuous templates.**

### 3.1 Prompting LLMs via Discrete Templates

As the most common type of prompts, discrete templates formulate the prompts using natural language. We designed two styles of discrete templates. One style is direct: directly asking whether an item is a serendipity. The second style is indirect, asking whether an item is unexpected and relevant, and then inferring whether it is a serendipity.

*3.1.1 Discrete Style 1: Direct.* This is the direct way to ask an LLM whether a candidate item is a serendipity to a user. As in Table 1 for this Style, the task description $I$ contains the definition of serendipity and the task requirement. In the demonstration set $D$, we expect the LLM learns from the representative examples to better understand the task, similar to the idea of providing some additional task-specific training instances in a supervised machine learning approach. We provide both serendipity (positive) and non-serendipity (negative) examples. For the input question $Q$, we include the user information $H_u$ and the candidate item information $i$. Specifically, $H_u$ is a series of item names that have been interacted by the user $u$. $i$ is just the candidate item name. In addition to $H_u$ and $i$, $Q$ also limits the answer format to be binary: "Yes" or "No" on whether the candidate item is a serendipity to this user, in order to prevent the LLM from being verbose or digressing.

*3.1.2 Discrete Style 2: Indirect.* This style prompts an LLM to breakdown the task of serendipity recommendations into two sub-tasks: judging whether an item is unexpected and then whether the item is relevant. Therefore, as in Table 1 for this Style, the task description $I$, in addition to defining serendipity, further provides the definitions of being unexpected and being relevant in the recommendation context. Accordingly, the demonstration set $D$ contains examples of serendipity items satisfying both conditions: being unexpected and being relevant. The input question $Q$ is similar to that of Discrete Style 1, but with an extra requirement to consider those two conditions when answering the question. Through this way, the serendipity recommendation task is converted into two question answering tasks.

The two styles above are to exploit both the direct and the indirect knowledge contained in an encyclopedia-like LLM through a series of predefined natural language prompts and the pre-specified possible answer words (yes or no). This is the core philosophy of the prompt learning paradigm, i.e., predicting an answer word from the LLM's vocabulary, as if the task-specific prompts had been inserted into the large corpus for training the LLM.

On the other hand, these manually designed templates, though with well-crafted statements, are obviously not exhaustive for all possible designs. Therefore we will use some virtual tokens to search for a few more template designs using the the continuous prompting approach.

**Table 1: Discrete prompt templates for serendipity recommendations**

| | | LLM Input |
|---|---|---|
| Discrete Style 1: Direct | Task Description $I$ | *Serendipity means an unexpected but relevant discovery to a user. Given a user's history, please answer "Yes" or "No" on whether a candidate item is a serendipity to the user.* |
| | Demonstration Set $D$ | *Here are some demonstrations:*
*Demo 1: Given a user $u_1$ with a history $H_{u_1}$ (a series of item names), $i_{cand}$ is a serendipity to this user u1.*
*...*
*Demo k: Given a user $u_k$ with a history $H_{u_k}$, $i'_{cand}$ is not a serendipity to this user $u_k$.* |
| | Input Question $Q$ | *Given a user $u$ with a history $H_u$, could you answer whether the candidate item $i$ is a serendipity to this user $u$?* |
| | | **Expected LLM Output** |
| | | *"Yes" or "No"* |
| | | **LLM Input** |
| Discrete Style 2: Indirect | Task Description $I$ | *Serendipity means an unexpected but relevant discovery to a user. Being serendipity means being both unexpected and relevant. In recommendation tasks, being unexpected means the items are unlikely to be recommended to a user given this user's history. Meanwhile, being relevant means the items are closely related to a user's history. Given a user's history, please answer "Yes" or "No" on whether a candidate item is a serendipity to the user. You need to consider both the unexpectedness and relevance aspects.* |
| | Demonstration Set $D$ | *Here are some demonstrations:*
*Demo 1: Given a user $u_1$ with a history $H_{u_1}$, $i_{cand}$ is both unexpected and relevant to this user $u_1$.*
*Therefore, it is a serendipity to this user $u_1$.*
*Demo 2: Given a user u2 with a history $H_{u_2}$, $i'_{cand}$ is relevant but not unexpected to this user $u_2$.*
*Therefore, it is not a serendipity to this user $u_2$.*
*...*
*Demo k: Given a user $u_k$ with a history $H_{u_k}$, $i''_{cand}$ is neither unexpected nor relevant to this user $u_k$.*
*Therefore, it is not a serendipity to this user $u_k$.* |
| | Input Question $Q$ | *Given a user $u$ with a history $H_u$, could you answer whether the candidate item $i$ is a serendipity to this user $u$? You need to consider both the unexpectedness and relevance aspects.* |
| | | **Expected LLM Output** |
| | | *"Yes" or "No"* |

## 3.2 Prompting LLMs via Continuous Templates

Different from the discrete prompts, continuous prompts use learnable tokens (vectors) in the task description $I$ and the input question $Q$. The two sets of learnable tokens can be generated by inputting the original natural words of $I$ and $Q$ through neural network layer(s) added in front of the LLM. The token generation process could be described as follows:

$$f(I, Q) \rightarrow [P_1^I]...[P_{n_1}^I]; [P_1^Q]...[P_{n_2}^Q] \qquad (3)$$

where $I$ and $Q$ are the natural words of the task description and the input question. $f(\cdot)$ is the added neural network layer(s), to map the natural words into two sets of tokens of $[P_1^I]...[P_{n_1}^I]$ and $[P_1^Q]...[P_{n_2}^Q]$ respectively. $n_1$ and $n_2$ are their numbers of tokens. Compared with the discrete prompts, the demonstration set $D$ is not included since $f(\cdot)$ can be trained by training instances, which essentially play the role of providing additional demonstrations to the LLM. Therefore the continuous prompts are all tokens, as in Table 2. Previous research shows that Transformers [25] are effective in mapping or encoding text into vectors. Therefore, in this paper, we selected the Transformers as $f(\cdot)$.

This way, **the continuous prompts provide more freedom by adding learnable tokens in search of the best prompts, although it may introduce some uncertainties and cost some explainability.** Similar to the discrete prompts, we designed two styles: direct and indirect, corresponding to the direct question

**Table 2: Continuous prompt templates for serendipity recommendations**

| LLM Input | |
|---|---|
| Task Description $I$ | $[P_1^I][P_2^I]...[P_{n_1}^I]$ |
| Input Question $Q$ | $[P_1^Q][P_2^Q]...[P_{n_2}^Q]$ |
| **Expected LLM Output** | |
| "Yes" or "No" | |

on serendipity and the indirect questions on unexpectedness and relevance.

*3.2.1 Continuous Style 1: Direct.* This Style trains $f(\cdot)$ to learn the tokens for the direct question on serendipity. The input of $f(\cdot)$, $I$, and $Q$, will be the same as Discrete Style 1 as in Table 1. During the learning process, the LLM is frozen, and only the parameters of $f(\cdot)$ are updated through the loss values between the model answers and the ground truths. We adopted the commonly used cross-entropy loss function to train $f(\cdot)$.

$$L_{seren}(\Theta) = -\frac{1}{|\mathcal{S}|} \sum_{(u,i) \in \mathcal{S}} y_{u,i} \, log(\hat{y}_{u,i}) \qquad (4)$$

where $\Theta$ is the set of the learnable parameters of $f(\cdot)$. $\mathcal{S}$ denotes the set of training instances for serendipity. $|\mathcal{S}|$ denotes the size of training instances. $\hat{y}_{u,i}$ is the predicted probability on "Yes" or "No". $y_{u,i}$ is the ground truth value.

*3.2.2 Continuous Style 2: Indirect.* In this Style, we did not directly learn $f(\cdot)$ for the prompt tokens for the question of serendipity. Instead, we first pre-trained $f(\cdot)$ with two pre-training tasks, and then fine-tuned it for the question on serendipity. The pre-train and fine-tune process is expected to learn better token representations of LLM prompts for serendipity recommendations. The two pre-training tasks are: learning a prompt representation to ask whether a candidate item is unexpected to a user; and learning a prompt representation to ask whether a candidate item is relevant to a user. Specifically, the $I$ and $Q$ for the first pre-training task are:

*I: "Unexpectedness means something that surprises somebody because the person is not expecting it. In recommendation tasks, being unexpected means the items are unlikely to be recommended to a user given this user's preferences. Given a user's history, please answer "Yes" or "No" on whether the candidate item is unexpected to the user."*

*Q: "Given a user with a history $H_u$, could you answer whether the candidate item $i$ is unexpected to this user $u$?"*

Similar to Continuous Style 1, during the pre-training process for this first pre-training task, the LLM is frozen, and only the parameters of $f(\cdot)$ are updated through the loss values between the model answers and the ground truths on unexpectedness. We adopted the commonly used cross-entropy loss function similar to Continuous Style 1. The $I$ and $Q$ for the second pre-training task are similar to the first pre-training task. The only change is from about unexpectedness to about relevance.

To pre-train $f(\cdot)$ with the two different pre-training tasks, a commonly used approach is sequentially training it with the two tasks. It may cause $f(\cdot)$ to "remember" only the second task and "forget" the first task [11, 24]. Therefore, we constructed a mixed two-task dataset to pre-train $f(\cdot)$ simultaneously. The mixed dataset sampled the training instances for the two pre-training tasks respectively and then mixed them in one dataset. That way, the pre-training process is able to strike a balance between the two tasks. Considering that two pre-training tasks may not contribute equally to the final vector representation, a sampling ratio $\frac{e}{l}$ was experimented to control the proportion of the two tasks in the mixed two-task dataset.

After being pre-trained, $f(\cdot)$ is expected to obtain some "prior knowledge" on serendipity. We further fine-tuned $f(\cdot)$ using the direct training instances on serendipity. The process is the same with the training task in Continuous Style 1.

Although continuous templates are more flexible compared to discrete ones, the quality of the generated tokens highly relies on the selection of $f(\cdot)$ and the training instances. More importantly, these tokens are randomly initialized, which may introduce some uncertainties and noises [29]. Therefore we further propose a hybrid prompt, which is a mixture of natural words and virtual tokens. We expect this type of prompt is able to provide stable input to the LLMs while controlling the level of uncertainties and noises.

## 3.3 Prompting LLMs via Hybrid Templates

In a hybrid prompt with both natural words and virtual tokens, we need to decide on what part(s) of the prompt should use natural words and what part(s) should use virtual tokens. **One principle we used is that if a part is information-rich, and natural words for it may not be sufficiently expressive, we will use virtual tokens. If a part needs to provide precise knowledge to the LLM without introducing any uncertainty, we will go with natural words.** Therefore, we selected the $H_u$ and $i$ elements in the input question $Q$ to be the virtual tokens, because they (a series of names) may not contain the rich information needed by the LLM to recommend serendipity. In contrast, the task description $I$ and the remaining part of $Q$ used natural words, since they reduce uncertainty and provide useful background knowledge. For the virtual tokens parts, we learned a mapping function $m(\cdot)$ to project the natural words of $H_u$ and $i$ to two sets of virtual tokens $[P_1^{H_u}]...[P_{n_3}^{H_u}]$ and $[P_1^i]...[P_{n_4}^i]$:

$$m(H_u, i) \rightarrow [P_1^{H_u}]...[P_{n_3}^{H_u}]; [P_1^i]...[P_{n_4}^i] \qquad (5)$$

Similar to the continuous prompts, we do not have the demonstration set $D$ in the prompts since the training instances for $m(\cdot)$ play the role of providing additional demonstrations. Similar to the discrete and continuous prompts, we also designed two styles for the hybrid prompt templates.

*3.3.1 Hybrid Style 1: Direct.* This style trains $m(\cdot)$ to generate token representations for a direct question of serendipity. As in Table 3 for this Style, the task description $I$ is the same with Discrete Style 1. The input question $Q$ is also the same with Discrete Style 1 except that we replaced $H_u$ and $i$ with virtual tokens.

To learn the optimal virtual tokens, we froze the LLM's parameters and only optimized $m(\cdot)$ by calculating the loss values between the model answers and the ground truths on serendipity. We used the cross-entropy loss function.

*3.3.2 Hybrid Style 2: Indirect.* Similar to Continuous Style 2, in this Style as in Table 3, we do not directly learn the tokens for the direct question of serendipity. Instead, we first pre-trained the tokens with those two pre-training tasks, and then fine-tuned them for the question of serendipity.

## 4 EXPERIMENTS

## 4.1 Construction of Ground Truths

In this paper, for continuous and hybrid prompts, we need the ground truth data to train, pre-train, or fine-tune the Transformer encoders added to the LLMs. For discrete prompts, we need the ground truth data to provide various positive and negative demonstrations. Specifically, we need three types of ground truth: serendipity, unexpectedness, and relevance.

For serendipity, we used *SerenLens* [4], an existing large-scale ground truth dataset on serendipity books, as the base dataset. We further converted this base into an instruction format that is compatible with the LLM prompt and output formats, in order to serve as the training instances for the Transformer encoders as well as demonstrations in discrete prompts. In total, we obtained 5,114 training instances.

For unexpectedness, we did not have any existing base dataset to convert. Therefore, we propose a computational approach to "calculate" the ground truth of unexpectedness, avoiding the tedious human labeling process. In Psychology, unexpectedness is defined as violation of expectation [19]. We propose a computational operationalization of this definition. Specifically, we first calculated the conditional likelihood of seeing an item given a user's history

**Table 3: Hybrid prompt templates for serendipity recommendations**

| | | LLM Input | |
|---|---|---|---|
| **Hybrid Style 1:** | Direct Serendipity Task Prompt | Task Description $I$ | *Serendipity means an unexpected but relevant discovery to a user. Given a user's history, please answer "Yes" or "No" on whether a candidate item is a serendipity to the user.* |
| | | Input Question $Q$ | *Given a user $u$ with a history $[P_1^{H_u}]...[P_{n3}^{H_u}]$, could you answer whether the candidate item $[P_1^i]...[P_{n4}^i]$ is a serendipity to this user $u$?* |
| | | **Expected LLM Output** | |
| | | *"Yes" or "No"* | |
| **Hybrid Style 2: Indirect** | Pre-Training Task 1 (Unexpectedness) Prompt | **LLM Input** | |
| | | Task Description $I$ | *Unexpectedness means something that surprises somebody because the person is not expecting it. In recommendation tasks, being unexpected means the items are unlikely to be recommended to a user given this user's preferences, usually represented by the user's history. Given a user's history, please answer "Yes" or "No" on whether the candidate item is unexpected to the user.* |
| | | Input Question $Q$ | *Given a user $u$ with a history $[P_1^{H_u}]...[P_{n3}^{H_u}]$, could you answer whether the candidate item $[P_1^i]...[P_{n4}^i]$ is unexpected to this user $u$?* |
| | | **Expected LLM Output** | |
| | | *"Yes" or "No"* | |
| | Pre-Training Task 2 (Relevance) Prompt | **LLM Input** | |
| | | Task Description $I$ | *Relevance means something being closely connected or related to the current topic of interest. In recommendation tasks, being relevant means the items are closely related to a user's preferences, usually represented by the user's history. Given a user's history, please answer "Yes" or "No" on whether the candidate item is relevant to the user.* |
| | | Input Question $Q$ | *Given a user $u$ with a history $[P_1^{H_u}]...[P_{n3}^{H_u}]$, could you answer whether the candidate item $[P_1^i]...[P_{n4}^i]$ is relevant to this user $u$?* |
| | | **Expected LLM Output** | |
| | | *"Yes" or "No"* | |
| | Fine-Tuning Task (Serendipity) Prompt | **LLM Input** | |
| | | Task Description $I$ | *Serendipity means an unexpected but relevant discovery to a user. Given a user's history, please answer "Yes" or "No" on whether a candidate item is a serendipity to the user.* |
| | | Input Question $Q$ | *Given a user $u$ with a history $[P_1^{H_u}]...[P_{n3}^{H_u}]$, could you answer whether the candidate item $[P_1^i]...[P_{n4}^i]$ is a serendipity to this user $u$?* |
| | | **Expected LLM Output** | |
| | | *"Yes" or "No"* | |

of interacted items. We then used a low level of such conditional likelihood as a high level of unexpectedness. Therefore, the level of unexpectedness of an item $i$ to a user with a history $H_u$ is calculated as:

$$unexp_{(u,i)} = -log\, p(i|H_u) \qquad (6)$$

where the negative sign is to indicate the opposite relationship between the cognitional likelihood and the level of unexpectedness. The logarithm function is to smooth the larger values. Using the Law of Total Probability, the conditional probability in Equation 6 could be rewritten as:

$$unexp_{(u,i)} = -log\, p(i|H_u) = -log \sum_{i^u \in H_u} p(i|i^u)p(i^u|H_u) \quad (7)$$

where $i^u$ is a user's historically interacted item in $H_u$, $p(i^u|H_u)$ is the occurring probability of $i^u$ in $H_u$, and $p(i|i^u)$ could be calculated as:

$$p(i|i^u) = \frac{n(i,i^u)}{\sum_{i \in I} n(i,i^u)} \qquad (8)$$

where the numerator $n(i,i^u)$ is the co-occurrence count for an item $i$ and $i^u$ in all users' histories. The denominator is the sum of such co-occurrence count over each item $i$ in the item set $I$. All of the

components on the right side of this Equation 7 could be calculated from a dataset. After calculating all the items' $unexp_{(u,i)}$ values for the user $u$ with the history $H_u$, we selected the items with the top values as the user's unexpected items (positive cases) and the items with the bottom values as the expected items (negative cases). We applied this approach to the Amazon Review Data [17] to calculate the level of unexpectedness between a book and a user. In total, we obtained 46,920 user-book pairs (the positive and negative pairs combined) and used them as the base to reformat according to the LLM prompt and output requirements to serve as the training instances or demonstrations.

For relevance, we used the Amazon Review Data [17] again and followed the common practice in the recommendation research community: the observed interaction between a user and a book establishes a relevance label for this user-book pair. Other user-book pairs with no observed interactions are deemed as irrelevance. We obtained more than 100 million cases (user-book pairs). We sampled them according to the sampling ratio ($\frac{e}{l}$) (mentioned in Section 3.2.2) with respect to the unexpectedness dataset. We then reformatted the sampled cases into training instances or demonstrations.

## 4.2 Backbone LLMs

We chose two open-source LLMs for the implementation of *Seren-Prompt*: Flan-T5 (the 11B version) and Llama 2 (the 13B version). They are two different representative open-source LLMs with strong performance on various tasks. Flan-T5 (the 11B version) is an enhanced version of T5 that has been fine-tuned in a series of tasks. It has comparable performance in many language tasks to much larger models, such as PaLM (the 62B version). Structurally, it has both Transformer encoders and decoders. On the other hand, Llama 2 (the 13B version) is an enhanced version of Llama 1 developed by Meta, with stronger performance. Structurally, it only has Transformer decoders. These two models are the two representative LLM structures.

## 4.3 Evaluation Metrics and Baseline Models

Since serendipity is relatively rare compared to non-serendipity in the ground truth datasets, we adopted a recall-based metric, Hit Ratio (HR). $HR_{seren}@k$ measures the proportion of times the serendipity item is retrieved in the top-k position (1 for yes and 0 otherwise). In order to take the rank information into consideration and assign higher weights on higher ranks, I propose another metric called Serendipity-Based Normalized Discounted Cumulative Gain ($NDCG_{seren}$) based on the well-known metric Normalized Discounted Cumulative Gain (NDCG). $NDCG_{seren}@k$ is calculated as:

$$NDCG_{seren}@k = \sum_{i=1}^{k} \frac{serendipity\ score(1\ or\ 0)}{log_2(i+1)} \quad (9)$$

For both $HR_{seren}@k$ and $NDCG_{seren}@k$, a higher value indicates a better performance.

To evaluate the performance of the series of *SerenPrompt*, we selected the following two groups of representative baseline recommendation models. The first group consists of one randomness-based method and four well-known deep learning recommendation models for serendipity: **RAND**, **DESR** [14], **PURS** [12], **SNPR** [28], and **SerenEnhance** [4].

The second group is two well-known deep learning recommendation models for relevance tasks: **SASRec** [9] and **BERT4Rec** [22]. Both of the two groups of models are state-of-the-art deep learning models published in top venues in recent years.

## 4.4 Experiment Setups

For the continuous and hybrid prompts, 80% of the data in *SerenLens* was used for training $f(\cdot)$ or $m(\cdot)$ and the rest 20% was for testing. Since discrete prompts do not have a training process, we directly used the testing set for evaluations. For all discrete prompts, we use 10 demonstrations with 5 positive ones and 5 negative ones. For Continuous Style 2 and Hybrid Style 2 prompts, we pre-trained the virtual tokens on the mixed two-task dataset sampled from the *UnexpectedBooks* and the *RelevantBooks* datasets. For all prompts, for each user in the test set, we held one serendipity item as the testing positive sample, and then paired it with 99 non-serendipity items that were randomly sampled from the dataset as the negative samples. We compared the LLM prompted by *SerenPrompt* with the baseline models using the metrics ($HR_{seren}@k$ and $NDCG_{seren}@k$). For all models, we adopted 5-fold cross-validation to evaluate the performance. We trained our models using the Adam optimizer. We

set the learning rate 0.001, the hidden dimension 128, the dropout rate 0.2, and the regularizer decay 0.001 for all the models. Other model-specific hyper-parameters either followed their original studies or were adjusted for the training performance in this study. We reported the results using the optimal hyper-parameter settings. In addition, for fair comparisons, we set the head number of the multi-head attention 2 for the models involving Transformers: SASRec, BERT4Rec, SNPR, SerenEnhance, and LLMs with *SerenPrompt*.

## 5 EXPERIMENT RESULTS

### 5.1 Hyperparameter Analysis

The core hyperparameters of *SerenPrompt* are the numbers of the virtual tokens (i.e., $n_1$, $n_2$, $n_3$, and $n_4$) and the sampling ratio ($\frac{e}{l}$) between the unexpectedness and relevance tasks in the mixed dataset in the pre-training stage for both the continuous and the hybrid prompts. We investigated the effects of changing these hyperparameters on the recommendation performance.

**Numbers of the virtual tokens.** Following the study of [29], we adopted a coarse hyperparameter training strategy, which makes $n_1 = n_2$ and $n_3 = n_4$. We chose $n_1$, $n_2$, $n_3$, and $n_4$ from the set of values {1,2,3,4,5} and explored the optimal settings with the best $HR_{seren}@10$. As shown in Figure 2a, for the continuous prompts, as $n_1$ or $n_2$ increases, the performance of *SerenPrompt* increases first and decreases then. When $n_1 = n_2 = 2$, $HR_{seren}@10$ reaches the highest values for both Flan-T5 and Llama 2. The results indicate that both too few and too many tokens will result in ineffective prompts. Too few tokens may lack sufficient task information, while too many tokens may suffer from noisy and ambiguous information.

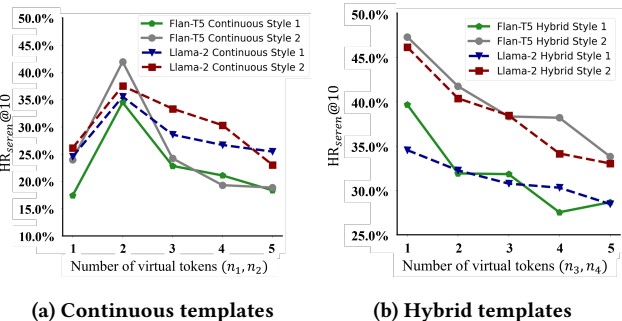

**(a) Continuous templates**  **(b) Hybrid templates**

**Figure 2: The recommendation performances of *SerenPrompt* using different numbers of virtual tokens**

For the hybrid prompts, as shown in Figure 2b, when $n_3 = n_4 = 1$, $HR_{seren}@10$ reaches the highest value for both Flan-T5 and Llama 2. That means only one token is sufficient to represent the user or the candidate item. We also observe that as the number of virtual tokens increases, the performance of the LLMs with *SerenPrompt* keeps decreasing. Compared with the continuous prompts, the hybrid ones require fewer virtual tokens. Therefore, in the following subsections, we will only report the results of *SerenPrompt* with $n_1 = n_2 = 2$ and $n_3 = n_4 = 1$ where applicable.

**Sampling ratio of the two-task dataset.** Using the optimal numbers of virtual tokens, we further investigated the effects of different $\frac{e}{l}$. As illustrated in Figure 3, Flan-T5 and Llama 2 have similar

**Table 4: The performance comparison of different *SerenPrompts* and baseline models on the serendipity recommendation task. The reported number is the average of 5 folds. The best results in each column are bolded and the second best results are underlined. ∗ denotes that our proposed model has statistically significant differences with all of the seven baseline models under a two-tailed t-test with p < 0.05.**

| | | $HR_{seren}@1$ | $HR_{seren}@5$ | $HR_{seren}@10$ | $NDCG_{seren}@5$ | $NDCG_{seren}@10$ |
|---|---|---|---|---|---|---|
| **Serendipity** | RAND | 1.15% | 4.51% | 9.16% | 0.028 | 0.043 |
| | DESR | 6.25% | 23.02% | 36.67% | 0.144 | 0.178 |
| | PURS | 5.76% | 22.60% | 32.20% | 0.139 | 0.170 |
| | SNPR | 7.46% | 24.09% | 38.81% | 0.149 | 0.192 |
| | SerenEnhance | 9.81% | 30.49% | 45.63% | 0.329 | 0.364 |
| **Relevance** | SASRec | 6.13% | 25.33% | 41.37% | 0.157 | 0.209 |
| | BBERT4Rec | 8.03% | 27.02% | 41.79% | 0.166 | 0.214 |
| **Flan-T5** | Discrete Style 1 | 4.65% | 17.83% | 31.17% | 0.105 | 0.142 |
| | Discrete Style 2 | 5.19% | 18.88% | 32.51% | 0.111 | 0.148 |
| | Continuous Style 1 | 6.03% | 28.96% | 34.48% | 0.159 | 0.186 |
| | Continuous Style 2 | 8.32% | 31.29% | 41.94% | 0.262 | 0.273 |
| | Hybrid Style 1 | 5.91% | 23.34% | 39.66% | 0.145 | 0.178 |
| | Hybrid Style 2 | **13.65%**∗ | **34.60%**∗ | **47.31%**∗ | **0.354**∗ | **0.398**∗ |
| **Llama 2** | Discrete Style 1 | 3.85% | 15.38% | 33.36% | 0.101 | 0.135 |
| | Discrete Style 2 | 4.16% | 16.18% | 34.46% | 0.107 | 0.145 |
| | Continuous Style 1 | 5.13% | 23.67% | 35.53% | 0.127 | 0.164 |
| | Continuous Style 2 | 8.64% | 26.43% | 37.50% | 0.188 | 0.218 |
| | Hybrid Style 1 | 5.68% | 17.69% | 34.54% | 0.115 | 0.152 |
| | Hybrid Style 2 | 11.38%∗ | 33.85%∗ | 46.15%∗ | 0.332∗ | 0.374∗ |

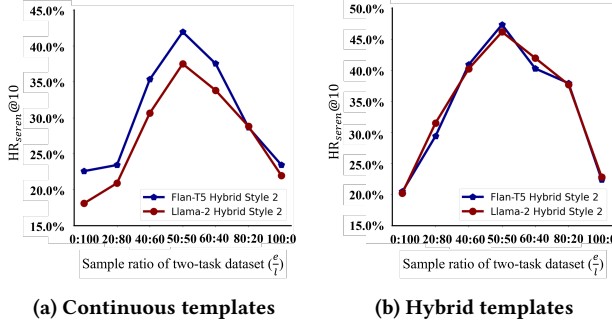

(a) Continuous templates          (b) Hybrid templates

**Figure 3: The recommendation performances of *SerenPrompt* using the two-task dataset with different sampling ratios**

trends for either continuous or hybrid prompts. When $\frac{e}{l} = \frac{50}{50}$, $HR_{seren}@10$ reaches the highest performance for both the prompts. The results indicate that a two-task dataset with an equal amount of training instances for the unexpectedness and relevance tasks is most effective to provide balanced "prior knowledge" on serendipity. In the following subsections, we will only report the results using $\frac{e}{l} = \frac{50}{50}$ where applicable.

## 5.2 Overall Performance Comparison

From Table 4, we know that the hybrid template with the pre-training and fine-tuning process (Hybrid Style 2: Indirect) achieves the best performance among all the baseline models and the other

types of *SerenPrompt*. In general, the LLMs prompted by the continuous templates and hybrid templates obtain a better performance than discrete templates. It suggests that compared to natural words, virtual tokens are more powerful in expressing a prompt for serendipity recommendations.

In addition, no matter which type of *SerenPrompt*, Style 2's performance is better than Style 1. The results prove the effectiveness of the decomposition of serendipity. Breaking down the direct question on serendipity into two sub-questions on unexpectedness and relevance is more helpful for the LLM to recommend serendipity.

It is interesting to note that prompting LLMs via discrete templates, which do not involve any model training, obtain a performance close to the other serendipity-oriented deep recommendation models (e.g., DESR, PURS, and SNPR). The results demonstrate the power of LLMs. They only need a task description and a few demonstrations to be on a par with the state-of-the-art baseline models. They are able to get around of the need of massive ground truth data. The potential offers many avenues for not only serendipity recommendations, but all kinds of recommendations in general.

## 5.3 Template Efficiency

Since the LLMs contain billions of parameters, the efficiency of prompting LLMs via *SerenPrompt* is another critical evaluation for serendipity recommendations. To compare the efficiency among different prompt templates, we calculated the average inference time of the LLM on 1 instance. We tested all the templates on the testing set of *SerenLens* dataset with 8 NVIDIA Tesla V100S GPUs. We kept all the hyperparameter settings the same for each template. The results are shown in Table 5.

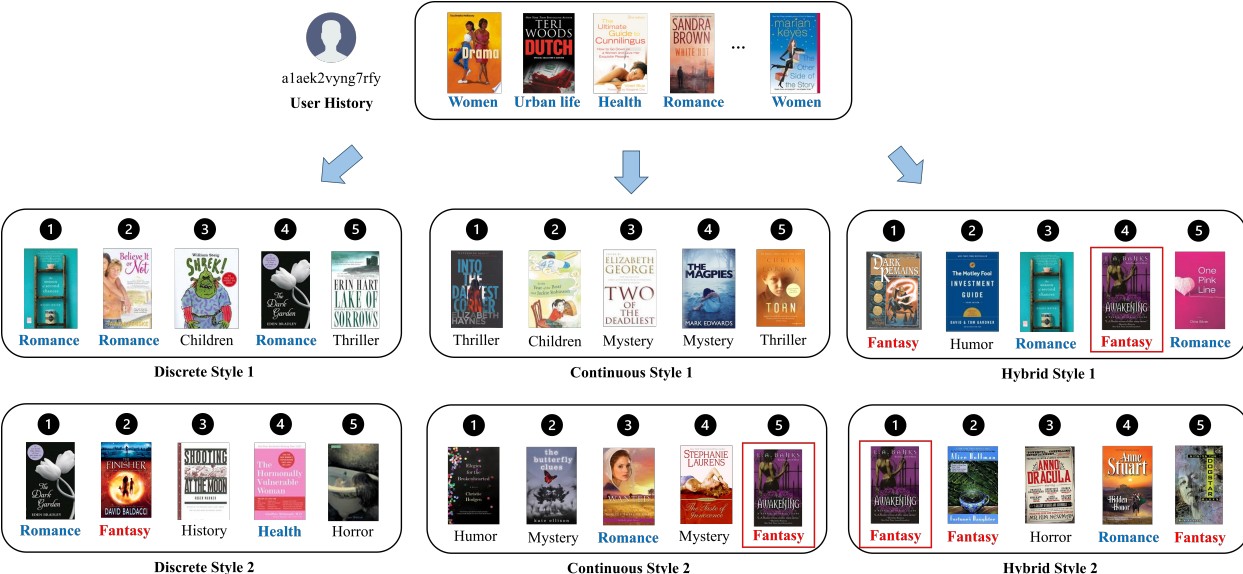

**Figure 4: Top-5 recommendation lists generated by different types and styles of prompts**

**Table 5: Prompt efficiency comparison**

|         |                     | HR$_{\text{seren}}$@10 | Inference Time in seconds |
|---------|---------------------|------------------------|---------------------------|
| **Flan-T5** | Discrete Style 1    | 31.17%                 | 5.57s                     |
|         | Discrete Style 2    | 32.51%                 | 8.90s                     |
|         | Continuous Style 1  | 34.48%                 | 0.38s                     |
|         | Continuous Style 2  | 41.94%                 | 0.40s                     |
|         | Hybrid Style 1      | 39.66%                 | 1.09s                     |
|         | Hybrid Style 2      | 47.31%                 | 2.85s                     |
| **Llama 2** | Discrete Style 1    | 33.36%                 | 20.14s                    |
|         | Discrete Style 2    | 34.46%                 | 21.68s                    |
|         | Continuous Style 1  | 35.53%                 | 4.16s                     |
|         | Continuous Style 2  | 37.50%                 | 3.63s                     |
|         | Hybrid Style 1      | 34.54%                 | 17.66s                    |
|         | Hybrid Style 2      | 46.15%                 | 21.51s                    |

We observe that the continuous templates for both Flan-T5 and Llama 2 have the least inference time while the discrete templates have the most. The results indicate that templates with more virtual tokens and fewer natural words improve the LLM's efficiency. In addition, the hybrid templates obtain the best recommendation performance on HR$_{\text{seren}}$@10 with relatively lower sacrifice to inference time, achieving a good compromise between the two conflicting goals: performance and efficiency.

### 5.4 A Case Study

To have an intuitive understanding of the model results, we selected an example user to showcase the recommendation results using different types and styles of prompts. As shown in Figure 4, there is a user interested in the books with topics of women, urban life, and romance according to her or his history. We also find that

this user had a serendipity experience on finding a book titled *The Awakening: A Vampire Huntress Legend* as in his or her written review:

"*I haven't picked up a good vampire novel in quite a while...I really didn't have high expectations for this novel or any others in this genre. What a surprise to discover that a Philadelphia Sistah has written a bona fide, nail-biting vampire novel that is equal if not better than Anne Rice, et al.*"

This book is a fantasy book with the romance and horror elements in it. It is not the usual type of this user. As shown in Figure 4, only the hybrid prompts with both styles and the Continuous Style 2 prompts were able to recommend this book in their top-5 recommendation list (as highlighted in the red box). The discrete prompts with both styles tend to recommend books more closely following this user's history, such as romance and health books. The continuous and the hybrid prompts are bolder and more deviating from the user's usual type.

## 6 CONCLUSIONS

In this paper, we investigated the potential of prompting LLMs to obtain serendipity recommendations. We designed three types of prompts: discrete, continuous, and hybrid, to investigate the expressiveness or effectiveness of natural words and virtual tokens used in prompts for serendipity recommendations. Meanwhile, for each type, we also designed two styles: direct and indirect, to investigate whether it is feasible to ask a direct question on serendipity or it is better to breakdown the direct question into two sub-questions. Extensive experiments have shown that the combination of the hybrid type and the indirect style achieves the best performance with relatively low sacrifice to computational efficiency, and outperforms all of the state-of-the-art baseline models.

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
