# OpenReview forum: "The Art of Asking: Prompting Large Language Models for Serendipity Recommendations"
_ACM.org/SIGIR/ICTIR/2024/Conference — ICTIR 2024_

### Official Review · Reviewer_86fz · 2024-05-02

**Rating:** 1
**Confidence:** 4

**Objective Part Of Review:**

The problem, method, and results of serendipity recommendations are clearly stated. Several writing problems:
- Figure 1 needs to be clarified. I cannot understand the notations' meaning solely with the Figure itself.
- Task Description $I$ is confusing with the set of items $I$.
- The $e$ and $l$ in the sampling ratio are not clearly explained.
- Does $n_1$, $n_2$, $n_3$, $n_4$ mean the virtual token number for **one discrete token OR the whole sequence**? It is evident that we cannot only use 1 or 2 tokens to represent the entire task description, question, history, etc, right?
- Typo of BBERT4Rec in Table 4.
---
The method can be polished to make the claim more reasonable, specifically:

- In Demonstration Set $D$, what is the relationship between the Demo users and the current user? Should we keep the Demo users the same as the current user for better prompting? Otherwise, the user information seems irrelevant. Also, what is the average input length of the discrete prompts since Demos may bring a lot of additional inputs?
- In Discrete Style 2: Indirect, did you try to separate unexpectedness and relevance aspects to ask LLM and then aggregate their Yes/No prediction in a post-processing manner? This can give us more detailed information about how LLM performs in these two aspects. Currently, all the aspects are integrated into one prompt, and we cannot know which contributes more to the final prediction.

---

The results about computational cost are needed:

- What are the total FLOPs for pre-training and fine-tuning your system? It is important to see these numbers and decide whether the improvements are worth such computation.
- The paper only compares the efficiency between their own models in 5.3, but I also want to see the inference time compared with baselines, especially SerenEnhance, whose performance is close to their proposed method.

**Subjective Part Of Review:**

The overall writing is clear and easy to understand. The methods of discrete and continuous prompts originate from the NLP community but are adapted to the recommendation field. They propose a novel method to calculate the ground truth of unexpectedness w/o human efforts. They can achieve state-of-the-art results in serendipity recommendations with hybrid prompting LLMs, but the efficiency of their model may be doubted. Here are my problems:
- Do you think using LoRA or Adapter to fine-tune LLMs efficiently can achieve better results than continuous prompts? Again, we still need the FLOPs of the proposed methods.
- Since the pre-training stage of the Transformer Encoder requires a lot of computation. I'd like to know if the paper can prove this encoder can generalize to more evaluation tasks, not only SerenLens.
- Do you think you can augment the item information with more description, not just the candidate item name, to achieve better performance even with the discrete prompts? It may induce longer input, but if it works, we may not need to train continuous prompts?
- Do you think using the GPT-4 level model to perform this task is feasible only with discrete prompts? If not, why?

---

### Official Review · Reviewer_ZRaG · 2024-05-12

**Rating:** 0
**Confidence:** 3

**Objective Part Of Review:**

This paper proposes SerenPrompt, containing discrete, continuous, and hybrid types, for the application of LLM to serendipitous recommendation. This approach incorporates the popular prompt tuning technique into the serendipitous recommendation domain for the first time.  The paper is well-organized and well-written, but may be missing some necessary citations.

**Subjective Part Of Review:**

**Strengths:**
- SerenPrompt design in the paper is an innovative way to handle serendipity recommendations using LLMs. It effectively combines the advantages of discrete, continuous, and hybrid prompts, enabling precise tuning for better recommendations.
- The authors made a notable contribution by integrating the concept of serendipity directly into the prompting process, which is a novel approach within the field of LLM-driven recommendation systems. This method might encourage more personalized and diverse user experiences.
- The paper’s experimental setup is rigorous, using standard metrics for evaluation, which helps in validating the effectiveness of the proposed methods against other state-of-the-art models.

**Weaknesses:**
- This paper tests SerenPrompt on two large language models which contain over 10 billion parameters. If possible, I'm curious if the authors could test this on a slightly smaller model (<10B), like llama3-8b? Do smaller models have that kind of power? Insights into how model scalability affects performance would be valuable, especially for practical implementations.
- Some research on prompt tuning need to be cited and featured in related work [1-3]. There has been some work on hybrid prompt tuning, and the innovation of this paper is to introduce the approach to the subfield of serendipity recommendation, which needs to be spelled out in more detail.
- Some comparative case studies would be helpful, illustrating the reasons for the failure of the baseline model and the proposed methodology for comparison would be good

[1] Gu, Yuxian, et al. "PPT: Pre-trained Prompt Tuning for Few-shot Learning." Proceedings of the 60th Annual Meeting of the Association for Computational Linguistics (Volume 1: Long Papers). 2022.

[2] Yang, Kexin, et al. "Tailor: A soft-prompt-based approach to attribute-based controlled text generation." Proceedings of the 61st Annual Meeting of the Association for Computational Linguistics (Volume 1: Long Papers). 2023.

[3] Zhang, Rui, et al. "Knowledge-Augmented Frame Semantic Parsing with Hybrid Prompt-Tuning." ICASSP 2023-2023 IEEE International Conference on Acoustics, Speech and Signal Processing (ICASSP). IEEE, 2023.

---

### Official Review · Reviewer_jNPZ · 2024-05-13

**Rating:** 0
**Confidence:** 4

**Objective Part Of Review:**

This paper proposes applying the LLM to the serendipity recommendation through prompt learning methods, and introduces a computational approach to assess serendipity, avoiding the cumbersome process of manual annotation. The paper is well-structured, and the experiments are fairly robust, but there are some issues that require clarification from the authors.

**Subjective Part Of Review:**

* The experimental setting is limited to the domain of the book. Is this because large language models possess richer knowledge in the domain of the book? The generalization ability of the proposed method needs further validation. Would it also hold advantages in e-commerce scenarios?
* Would this method yield the same conclusions on smaller-scale models?
* The novelty of this paper requires further explanation. I'm curious about the necessity of incorporating prompt learning and large models, considering potential efficiency sacrifices.
* Would user-item interaction histories pose challenges with long-text inputs? If so, how would these be addressed?
* In comparison with the primary baseline model in terms of efficiency, I'd like to know whether the efficiency trade-off is acceptable alongside the improvements.

---

### Official Review · Reviewer_czj9 · 2024-05-21

**Rating:** 1
**Confidence:** 4

**Objective Part Of Review:**

In this paper, the authors investigated the potential of prompting LLMs to obtain serendipity recommendations. Three types of prompts: discrete, continuous, and hybrid, are designed to investigate the expressiveness or effectiveness of natural words and virtual tokens used in prompts for serendipity recommendations. Extensive experiments have demonstrated the effectiveness of SerenPrompt in generating serendipity recommendations, compared to the state-of-the-art models.

**Subjective Part Of Review:**

Strengths:

1. The paper introduces an approach to leveraging Large Language Models for generating serendipity recommendations, a relatively new method in recommender systems.

2. The introduction of SerenPrompt, with variations in discrete, continuous, and hybrid forms, adds significant depth and robustness to the study.

3. Extensive experiments were conducted that not only demonstrate the effectiveness of SerenPrompts but also compare them against state-of-the-art models, achieving excellent performance.

4. The paper provides clear definitions and decomposes the concept of serendipity.


Weaknesses:

1. The heavy use of technical jargon could be reduced to enhance readability.

2. While the hybrid approach shows promising results, the computational costs and practicality of deploying such LLM-based systems in real-world scenarios should be thoroughly discussed.

3. The focus is primarily on serendipity recommendations without discussing how these findings might generalize to other types of recommendation systems or domains.

4. There is a lack of discussion on potential ethical issues and biases that might arise from serendipity recommendations, especially given the subjective nature of what might be considered serendipitous.

Three questions for the authors:

First, If you change the large language model that you use, will the experimental results change completely?

Second, all prompt methods so far are somewhat randomized since changing just a few words may give different generated results. How are the authors going to address this issue in this paper?

Third, it looks like that these tailored prompt methods will inevitably produce better results. Is it because of the power of LLM itself? Can the authors explain why?

---

### Meta-Review · Area_Chair_fL4p · 2024-05-31

**Recommendation:** Accept (Oral)
**Confidence:** 5

**Metareview:**

The paper compares different ways to obtain serendipitous recommendations, via different forms of prompting, and via vector similarity.

  Relevance:

* The paper is very relevant for conference topics.



Soundness:

* The paper is a good idea. A bit straight-forward, but yes, this is how I would do it too.

* The empirical evaluation is thorough. It only uses one data set SerenLens on book recommendation, but I assume that serendipity datasets are hard to come by.



Innovation:

* I am not sure what to take from this paper other than, "yup, it works". But maybe this is exactly what we want in a paper.


Several reviewers pointed out smaller issues that can be easily addressed.

Overall, I think the paper is worthwhile accepting.